# A METHOD FOR IDENTIFYING CAUSALITY IN THE RE-SPONSE OF NONLINEAR DYNAMICAL SYSTEMS

## ABSTRACT

Predicting the response of nonlinear dynamical systems subject to random, broadband excitation is important across a range of scientific disciplines, such as structural dynamics and neuroscience. Building data-driven models requires experimental measurements of the system input and output, but it can be difficult to determine whether inaccuracies in the model stem from modelling errors or noise. Therefore there is a need to determine the maximum component of the output that could theoretically be predicted using the input, if an improved model was to be developed through the investment of resources. This paper presents a novel method to identify the component of the output that could potentially be modelled, and quantify the level of noise in the output, as a function of frequency. The method uses input-output measurements and an available, but approximate, model of the system. A trainable, frequency dependent parameter balances an output prediction generated by the model with noisy measurements of the output to predict the input to the system. This parameter is utilised to estimate the noise level and then calculate a nonlinear coherence metric as a measure of causality or predictability from the input. There are currently no solutions to this problem in the absence of an accurate benchmark model.

## 1 INTRODUCTION

Nonlinear dynamical systems are observed across a wide range of scientific disciplines (Mosekilde, 1997), from structural dynamics (Nayfeh & Mook, 1979; Worden & Tomlinson, 2001) and electrical engineering (Kyamakya et al., 2013), to neuroscience (Corinto & Torcini, 2019) and economics (Bischi, 2010). Modelling their behaviour offers meaningful insights, enables predictions, and allows for control to generate desirable outcomes.

Accurate models generally involve experimental measurements, e.g. to carry out model updating for physics-based models or to train data-driven models. However, it can be difficult to determine whether the prediction errors are due to additional noise or inaccuracies in the model. Consider a nonlinear dynamical system subject to broadband random excitation, $x(t)$, which is especially prevalent in structural dynamics. The output of the system is measured to be $y_n(t)$, where

$$y_n(t) = y(t) + \epsilon_n(t) = \mathcal{M}\{x(\tau)_{\tau \leq t}\} + \epsilon_n(t) . \tag{1}$$

Here $\mathcal{M}\{\cdot\}$ represents the true causal dynamical system, $\epsilon_n(t)$ is additive noise and $y(t)$ is the component of $y_n(t)$ that is caused by $x(t)$. The noise is assumed to be due to uncoupled, unmeasured signal sources rather than measurement noise, hence there is no noise on the input. A forward model can be used to make an imperfect prediction $y_z(t)$, where

$$y_z(t) = \mathcal{G}\{x(t)\} = \mathcal{M}\{x(\tau)_{\tau \leq t}\} + \epsilon_z(t) . \tag{2}$$

Here $\epsilon_z(t)$ denotes the model errors which are caused by the inability of the model, $\mathcal{G}\{\cdot\}$, to fully capture the functional $\mathcal{M}\{\cdot\}$. Determining the component of $y_n(t)$ caused by $x(t)$ is important for applications which depend on the ability to predict the best possible estimate of $y_n(t)$ from $x$. However, this is challenging because it is difficult to distinguish between $\epsilon_n(t)$ and $\epsilon_z(t)$.

The input-output causal relationship can be quantified, as a function of frequency, using a nonlinear coherence metric between $x(t)$ and $y_n(t)$. Similar to correlation, it is a metric between

0 and 1 of the component of $y_n(t)$ that can be predicted using $x(t)$, as a function of frequency. In this context, it is a measure of causality because $x(t)$ is the signal which directly excites the system to generate $y$. This is also equal to the linear coherence, $\gamma^2_{YY_n}(f)$ (Kay, 1993), between $y(t)$ and $y_n(t)$ which cannot be calculated directly due to the unavailability of $y(t)$. Only a lower bound on the nonlinear coherence can be estimated using a model prediction, $y_z(t)$. Determining the true nonlinear coherence is challenging because it is difficult to distinguish between $\epsilon_n(t)$ and $\epsilon_z(t)$, which both appear as random noise. When the input to the system can be controlled, it is possible to estimate the contribution of $y(t)$ through repeat measurements (see Appendix A). However, when the input cannot be controlled, there are currently no methods that can infer the nonlinear coherence, $\gamma^2_{YY_n}(f)$, without a high-accuracy model of $\mathcal{M}\{\cdot\}$, which can require large models and lots of data to compute. Ideally, $\gamma^2_{YY_n}(f)$ could be estimated using a small quantity of data to determine the feasibility of a project before investing in data collection and computational resources.

Despite its wide ranging applications, this work is specifically motivated by feedforward active noise reduction (ANR), where $x$ is used as a reference signal to reduce the target signal, $y_n(t)$. In order to reduce the unwanted noise, a model must be developed to accurately predict $y_n(t)$ using $x(t)$. The maximum noise reduction that can be achieved is quantified by the causal relationship between $x(t)$ and $y_n(t)$. For linear systems, such as those in sound cancelling earbuds and headsets, this is quantified using the linear coherence metric, and the level of noise reduction that can be achieved, in decibels, is equal to $\Gamma_{XY}(f) = 10\log_{10}\left(1 - \gamma^2_{XY}\right)$. However, ANR has recently begun implementation in cars, where the transmission pathway from $x$ to $y_n$ is highly nonlinear (De Brett, 2020; Oh et al., 2015). For nonlinear systems such as this, the nonlinear coherence would be used to quantify the level of noise reduction that is possible. This is crucial in evaluating the benefits of investing resources into developing a high fidelity model of the car to enhance ANR performance. It is not currently possible to calculate the nonlinear coherence without investing the necessary resources into building the best model possible.

This work aims to estimate the nonlinear coherence between $x(t)$ and $y_n(t)$ for a particular class of nonlinear dynamical systems, with relatively small quantities of data and an incomplete model. These systems can be described by the general class of second order ordinary differential equation (ODE) $\mathcal{N}(y, \dot{y}, \ddot{y}) = x$, where $\mathcal{N}(y, \dot{y}, \ddot{y})$ represents an arbitrary nonlinear function. The most common form of ODE of this form seen in practice is:

$$\frac{1}{\omega_n^2}\ddot{y} + \frac{2\zeta}{\omega_n}\dot{y} + y + \mathcal{N}(y, \dot{y}) = x \ . \tag{3}$$

Providing measurements are taken across nonlinearities (which are often localised), this can be extended to systems with multiple degrees-of-freedom (extension to multiple degrees-of-freedom is part of ongoing work). Nonlinear dynamical systems of this type are common in many scientific disciplines and there has been significant work in modelling their response using data-based machine learning methods (Wang et al., 2016; Liang et al., 2005; Worden et al., 2018; Massingham et al., 2024). However, it is extremely difficult using feedforward modelling techniques (e.g. convolutional neural networks) due to the exponentially growing number of nonlinear monomial terms which contribute to the output. This numerical challenge is captured by the Volterra series perspective (Volterra, 1959). It is especially challenging for systems with long memory and for random broadband inputs. In this paper, the structure of Eq. 3 is exploited to estimate the nonlinear coherence for systems of this type, without a perfect model of the system. This enables the calculation of the causal component of a system response, which can be used to guide the investment of time and money into high performance model development. To the best of the author's knowledge, there are currently no other methods that achieve this. The key contributions of the work are as follows:

- A new strategy is introduced for estimating the **maximum possible performance** of time series prediction algorithms in the presence of noise.

- A novel method is presented for calculating the **nonlinear coherence** between input and output data for a class of nonlinear ODEs subject to random input, in the presence of noise.

- The robustness of the method is demonstrated on three simulated nonlinear dynamical systems which are **widely applicable** across several scientific disciplines.

- The applicability of the method is demonstrated on an **experimental** dataset with a strong nonlinearity.

## 2   RELATED WORK

The presented method for inferring the maximum performance of feedforward models and calculating the level of noise present is novel, and there are no similar methods in the literature. However, there are three broad areas of research in the literature that overlap with, and contribute to, the goals of this paper.

**Kalman Filter** The Kalman filter (Kalman, 1960), and its extensions, aim to estimate the true state of a system by optimally combining the predictions of a state space model with noisy measurements. A comparison of the model error and noise level is used to balance the two signals optimally. This relies on knowledge of at least one of the model error or noise variance because the true system state is not available for hyperparameter tuning. In this paper, the primary aim is to estimate the level of noise and model error, but a similar principle is utilised.

**Scaling Laws** Estimating scaling laws for various machine learning tasks has received significant attention in recent years (Hestness et al., 2017; 2019; Henighan et al., 2020), with the aim of forecasting how the error of various models will improve as more data is collected and larger models with more parameters are built. However, this approach is limited, as it relies on using the performance of a model trained on a small quantity of data to extrapolate and predict the performance of a bigger model trained using large amounts of data. Extrapolation is difficult, as it assumes there is no noise; the loss will plateau at the noise level. It is therefore difficult to evaluate the maximum performance unless the model is already close to the maximum, at which point increasing the model size and number of training examples will not reduce the prediction error. For the method presented in this paper, even with a poor forward model, it is possible to estimate the maximal predictive performance of a feedforward model by identifying the level of noise.

**Learning Input Noise Levels** As discussed above, if there is noise added to the output, it is difficult to determine the noise level because it is difficult to differentiate between unmodelled components of the system response and noise. However, if there is noise in the input, there are methods for inferring the true input and hence the input noise level. The Error-in-Variable approach can be used to infer the input to a machine learning algorithm and hence infer the noise level (Van Gorp J, 2000; Seghouane & Fleury, 2001). Therefore, with the method presented in this paper, by inverting the problem and trying to predict the input to the system using the output, the output noise could be determined. The issue with applying this to time series data is it requires inferring the true system response for potentially millions of time samples, and so the inference is extremely high dimensional. This paper proposes an alternative approach to this problem, which is well suited to time series applications.

Other methods have been presented in the literature that aim to identify a 'nonlinear' coherence. However, these methods generally rely on a perfect forward model of the system and so model error and additive noise are combined in any estimation (Worden et al., 2018). Our approach is distinct in that only an approximate forward model is needed.

## 3   METHODOLOGY

### 3.1   ARCHITECTURE

Consider the signals $x(t)$, $y_n(t)$, and $y_z(t)$. Here $x(t)$ is the input to the system, $y_n(t)$ is a noisy observation and $y_z(t)$ is the prediction of an imperfect forward model. The selection of the forward model is discussed in Section 3.3. The noise is represented by $\epsilon_n(t)$, which is assumed to be zero mean but can otherwise be of an arbitrary spectrum, and the true output is denoted $y(t)$. The time series data is split into $N$ frames (input-output pairs), each of length $M$. The data is split into training and validation datasets, with $N_T$ and $N_V$ frames respectively. At each frequency, $Y^{(i)}(f)$, $Y_n^{(i)}(f)$ and $Y_z^{(i)}(f)$ represent the Fourier transforms of the true, measured, and predicted versions of $y$ at frequency $f$ for frame $i$. The terms $\mathcal{E}_n^{(i)}(f)$ and $\mathcal{E}_z^{(i)}(f)$ represent the zero mean noise and model error respectively at frequency $f$ in frame $i$:

$$Y_n^{(i)}(f) = Y^{(i)}(f) + \mathcal{E}_n^{(i)}(f) \qquad\qquad Y_z^{(i)}(f) = Y^{(i)}(f) + \mathcal{E}_z^{(i)}(f) \,, \qquad (4)$$

where $\mathcal{E}_n^{(i)}(f)$ is assumed to be uncorrelated with $Y^{(i)}(f)$, whereas $\mathcal{E}_z^{(i)}(f)$ is correlated with $Y^{(i)}(f)$. The explicit dependence on $f$ and $t$ will now be dropped for compactness: lowercase denotes the time domain and uppercase denotes the frequency domain.

A key observation of the structure of Eq. 3 is that whilst it is difficult to learn a mapping from $x$ to $y$ in the 'forward' direction, it is relatively simple to map from $y$ to $x$ in the 'reverse' direction, even for highly nonlinear systems with long memory. This is because there is no nonlinear mixing of terms in time in the reverse direction, and so there is not an infinite Volterra series (Volterra, 1959) as there is in the forward direction. Therefore, even for highly nonlinear systems with long memory, there is a relatively simple mapping from $y$ back to $x$. This generalises to systems with multiple degrees-of-freedom, providing the nonlinear degrees-of-freedom are measured and so the memory of the reverse mapping is linear and hence straightforward to learn. In addition, it could be applied to dynamical systems beyond those with the structure described by Eq. 3, providing the memory of the reverse mapping is not long and nonlinear.

In the simplest case, this indicates whether there is any noise present, because if $x$ can be perfectly reproduced using $y_n$, there is no noise. On the other hand, if $x$ can not be perfectly reproduced using $y_n$, assuming a simple mapping exists, this indicates there is a component of $y_n$ that has not been caused by $x$, i.e. noise. However, quantifying the level of noise present is challenging. Kalman filters (Kalman, 1960) estimate the state of a system by balancing model predictions with noisy measurements based on the relative size of their errors. By defining a metric to quantify the quality of state estimation, the level of noise and model error can be inferred by optimising this metric. In this work, the input, $x$, is used as a reference signal to quantify the quality of the estimation of $y$, which is estimated by combining $y_z$ and $y_n$. Therefore, by optimising the estimation of $x$, the noise level and model error can be inferred. Figure 1 illustrates the proposed architecture.

In the frequency domain, consider a linear combination of $Y_n^{(i)}$ and $Y_z^{(i)}$ at each frequency:

$$\hat{Y}^{(i)} = Y_n^{(i)} K + Y_z^{(i)} \left(1 - K\right) , \tag{5}$$

where $0 \leq K(f) \leq 1$. As shown below, $K$ encapsulates information regarding the noise level and model inaccuracy in order to optimise the estimation of $y$ and therefore $x$. In the time domain, this is then used to train a machine learning model, alongside the parameter $K$, to predict $x$ by minimising the cost function:

$$\mathcal{L}^T = (1 - \lambda)\mathcal{L}_x^T + \lambda\mathcal{L}_y^T . \tag{6}$$

The losses $\mathcal{L}_x^T$ and $\mathcal{L}_y^T$ are calculated as

$$\mathcal{L}_x^T = \sum_{i=1}^{N^T} \frac{||\mathcal{F}_\theta\left(\hat{y}^{(i)}\right) - x^{(i)}||^2}{\sigma_x^2} \qquad \mathcal{L}_y^T = \sum_{i=1}^{N^T} \frac{||\hat{y}^{(i)} - y_n^{(i)}||^2}{\sigma_y^2} , \tag{7}$$

where $\mathcal{F}_\theta$ has memory and represents the trained model with parameters $\theta$ and the superscript $T$ indicates the loss is calculated using the training dataset. The variances of the random signals $x$ and $y_n$ are equal to $\sigma_x^2$ and $\sigma_y^2$ respectively, which are calculated as

$$\sigma_x^2 = \frac{1}{N} \sum_{i=1}^{N} ||x^{(i)}||^2 \qquad \sigma_y^2 = \frac{1}{N} \sum_{i=1}^{N} ||y_n^{(i)}||^2 . \tag{8}$$

Each term in the loss is normalised using the variance of each signal. The constant $\lambda$ plays an important role, as it balances the two terms in the loss function. The loss $\mathcal{L}_x$ drives the architecture towards predicting $x$ accurately, whereas $\mathcal{L}_y$ drives the architecture towards $\hat{y} = y_n$. This affects the optimal $\hat{y}$, and hence the optimal $K$. When $\lambda$ is set to 0 then the only term contributing to the loss function is $||\mathcal{F}_\theta\left(\hat{y}^{(i)}\right) - x^{(i)}||^2$ and it is hypothesised that the optimal value of $K$ will also yield the optimal estimate of the noise-free output $Y$ in order to estimate $x$ optimally. This optimum $K$ corresponds to minimising:

$$J = E\left[|\hat{Y} - Y|^2\right]. \tag{9}$$

Minimising this expectation with respect to $K$, by setting $\frac{\partial J}{\partial K} = 0$, yields:

$$K = \frac{1}{1 + \frac{E[\mathcal{E}_n \mathcal{E}_n^*]}{E[\mathcal{E}_z \mathcal{E}_z^*]}} . \tag{10}$$

See the Appendix B.1 for more details. This is the true optimum value of $K$ for a given frequency, but it cannot be computed directly because $\mathcal{E}_n$ and $\mathcal{E}_z$ are unknown. However, under the hypothesis that $\lambda = 0$ yields the true optimum for $K$, then it is expected that $K$ will converge towards this value in order to minimise $\mathcal{L}$. This is because the in order to optimally predict $x$ using $y_n$, as much noise must be removed as possible, indicating that $\hat{y}$ should be as close to $y$ as possible. Therefore if the parameters of the model, $\theta$ and $K$, are trained simultaneously, the architecture will implicitly estimate the ratio of errors in order to optimally combine the two signals. This is used to calculate the nonlinear coherence in Section 3.2.

In practice, $\lambda = 0$ does not yield the optimal $K$ and the role of $\lambda$ will be discussed in detail in Section 3.4. In this paper, $\mathcal{F}_\theta$ is a 1D convolutional neural network (CNN). Due to the simplicity of the reverse mapping from $y$ to $x$, a relatively small CNN can be constructed, even when the system memory from $x$ to $y$ is relatively long. For the examples below, a CNN with 5 layers, a kernel width of 7 and 5 features was used. This network has 778 parameters and was applied in all the example test cases below.

The optimum value of $K$ could be estimated independently at each frequency, which would result in $M$ parameters, where $M$ is the frame length for the Fourier transform. However, it will vary with some structure across the frequency domain, and so could be represented using a smaller number of parameters, which will improve its learnability. There are many possible ways to represent $K$, such as using a neural network, gaussian process or a kernel based method. A linear piece-wise approximation is chosen here for simplicity, where $M_c$ uniformly spaced points are learnt: $K_c(f_j)$ for $j = 1 : \frac{M}{M_c} : M$. Linear interpolation is then used between these points to obtain $K(f_i)$ for $i = 1 : 1 : M$. The resulting function is passed through the sigmoid function in order to constrain it to lie between 0 and 1.

### 3.2 NONLINEAR COHERENCE

The nonlinear coherence between $x$ and $y_n$ is equal to the linear coherence between $y$ and $y_n$:

$$\gamma^2_{YY_n} = \frac{|S_{YY_n}|^2}{S_{YY}S_{Y_nY_n}} \ , \tag{11}$$

where $S_{AB} = E[AB^*]$ represents the cross spectral density between signals $A$ and $B$ at frequency $f$. Assuming $\mathcal{E}_n$ and $Y$ are uncorrelated, the following can be written.

$$S_{YY_n} = E[YY_n^*] = E[YY^*] \tag{12}$$
$$S_{Y_nY_n} = E[Y_nY_n^*] \tag{13}$$
$$S_{YY} = E[YY^*] \ . \tag{14}$$

Therefore

$$\gamma^2_{YY_n} = \frac{E[YY^*]}{E[Y_nY_n^*]} \ . \tag{15}$$

Note that $E[Y_nY_n^*]$ is not simplified further because this quantity can be measured from data. An estimate of $E[YY^*]$ is required. The estimation of $K$, using the architecture described in Section

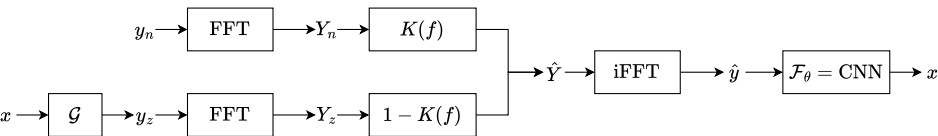

Figure 1: Architecture. The signals $x$ and $y_n$ are measured, whilst $y_z$ is generated from $x$ using an available model of the system, $\mathcal{G}$. The fast Fourier transform (FFT) of the signals $y_z$ and $y_n$ is taken, before the two signals are combined at each frequency point using the parameter $K(f)$. The inverse fast Fourier transform (iFFT) is then applied to the resultant signal, which is mapped to $x$ using a CNN. The parameters of the CNN, $\theta$, and the parameter $K(f)$ are trained simultaneously.

3.1, can be used to reverse engineer an estimation of $E\left[YY^*\right]$. Rearranging Eq. 10, the ratio between the error in each signal is given as

$$C \equiv \frac{E\left[\mathcal{E}_z \mathcal{E}_z^*\right]}{E\left[\mathcal{E}_n \mathcal{E}_n^*\right]} = \frac{K}{1-K} \; . \tag{16}$$

Therefore

$$CE\left[\mathcal{E}_n \mathcal{E}_n^*\right] = E\left[\mathcal{E}_z \mathcal{E}_z^*\right] \tag{17}$$

$$= E\left[(Y-Y_z)(Y-Y_z)^*\right] \tag{18}$$

$$= E\left[YY^*\right] - 2\Re\{E\left[YY_z^*\right]\} + E\left[Y_z Y_z^*\right] \tag{19}$$

In addition to this equation, three quantities can be calculated directly from the data: $E\left[Y_n Y_n^*\right]$, $E\left[Y_z Y_z^*\right]$ and $E\left[Y_z Y_n^*\right]$. Noting that $\mathcal{E}_z$ is correlated with $Y$ because it includes model error, two additional equations can be written:

$$E\left[Y_n Y_n^*\right] = E\left[YY^*\right] + E\left[\mathcal{E}_n \mathcal{E}_n^*\right] \tag{20}$$

$$E\left[Y_z Y_n^*\right] = E\left[Y_z Y^*\right] \; . \tag{21}$$

Combining Eq. 19 and Eq. 20 to eliminate $E\left[\mathcal{E}_n \mathcal{E}_n^*\right]$ gives the following.

$$C\left(E\left[Y_n Y_n^*\right] - E\left[YY^*\right]\right) = E\left[YY^*\right] - 2\Re\{E\left[YY_z^*\right]\} + E\left[Y_z Y_z^*\right] \; . \tag{22}$$

Solving for $E\left[YY^*\right]$ using Eq. 21 and Eq. 22 gives

$$E\left[YY^*\right] \approx \frac{CE\left[Y_n Y_n^*\right] - E\left[Y_z Y_z^*\right] + 2\Re\{E\left[Y_n Y_z^*\right]\}}{C+1} \; , \tag{23}$$

which can be calculated directly from the data using the inferred value of $K$ together with Eq. 16. This is then substituted into Eq. 15 to estimate the nonlinear coherence.

### 3.3 GENERATING THE FORWARD PREDICTION

The architecture described in Section 3.1 requires a forward prediction of $y$, generated using a model $\mathcal{G}$. This may be the best forward prediction currently available but it can be a simpler model too. In this paper, the prediction of a temporal CNN trained on the available data is used. This is an arbitrary choice and any forward model could be used because the presented method uses the forward prediction to estimate an improved estimation of the nonlinear coherence. The better the available forward model, the better the estimation of the nonlinear coherence; this is demonstrated in the simulations below by applying the CNN to systems with varying strengths of nonlinearity. It has been found that it is preferable to use a more complex model, such as a CNN, so that the forward prediction cannot be easily mapped back to the input and the architecture drives towards $\hat{y} = y$, as discussed in Section 3.4.

### 3.4 ROLE OF $\lambda$

The solution for $K$ found by the architecture described in Section 3.1 with $\lambda$ set to 0 is usually smaller than the true value of $K$ across the entire frequency range. It is hypothesised that this is because $y_z$ is generated directly from $x$, and so there is often a mapping directly from $y_z$ back to $x$, rather than by inferring $y$ first. The term $\lambda$ was therefore included to add some bias to drive the architecture towards inferring $y$; the resulting value of $K$ was used to infer the nonlinear coherence. Its value is determined using the validation dataset. As $\lambda$ increases, initially $\mathcal{L}_x^V$ remains constant, before beginning to increase rapidly. The best estimation of the nonlinear coherence is found at the point when $\mathcal{L}_x^V$ begins to increase rapidly. In order to identify this point, $\lambda$ was increased incrementally and the model was trained for 100 epochs at each step. When the normalised loss $\mathcal{L}_x^V$, averaged over the past 100 epochs, increased by 0.001 above the minimum value, the previous value of $\lambda$ was chosen. If $\mathcal{L}_x^V$ does not increase, as is the case for low levels of noise, $\lambda = 0.99$ is used. Example plots of $\lambda$ against $\mathcal{L}_x^V$ from the tests in Sections 4 and 5 can be found in Appendix D.

### 3.5 TRAINING

Both the neural network weights and $K$ are trained using the Pytorch infrastructure (Paszke et al., 2019). A learning rate of 0.01 was used with the Adam optimisation algorithm. For $\lambda = 0.001$, the architecture was trained for 2000 epochs. Then, for each subsequent value of $\lambda$, the algorithm was trained for 100 epochs. The full details of the implementation and training procedure are available on github (see the reproducibility statement).

## 4    SIMULATIONS

The method was evaluated on three simulated systems, each with different functional forms of nonlinearities in order to demonstrate the applicability and usefulness of the method. For each system, the robustness of the method was tested for different levels of noise. The spectrum of the noise was bandlimited and effectively flat around the resonant peak. The noise level can be quantified by the nonlinear coherence between $x$ and $y_n$, which is represented equivalently here as $\mathrm{Co}(Y, Y_n)$: values close to 1.0 indicate low noise and values close to 0 indicate high noise. In each case, the nonlinearity of the test case is quantified by the normalised mean squared error between the true response and the linearised response. The method presented in this paper was used to predict the nonlinear coherence between $x$ and $y_n$, using the $K(f)$ obtained from the architecture defined in Section 3.1, and the derivation of $\gamma_{YY_n}^2(f)$ in Section 3.2. The predictions were made using just 10 frames of data.

To benchmark the performance of the algorithm, the predictions of the nonlinear coherence are compared against those of the forward model, denoted $\mathrm{Co}(Y_z, Y_n)$, and a linearisation of the system, denoted $\mathrm{Co}(X, Y_n)$. The forward model, described in Section 3.3, was chosen to be a 1D temporal convolutional neural network with dilations, as described by van den Oord et al. (2016). The prediction of this forward model was used as $y_z$ to demonstrate the improved estimation of the nonlinear coherence that can be calculated using the method presented in this paper. The generated $Y_z$ is therefore used both to provide a baseline and also to estimate the nonlinear coherence. It is likely that better forward models could be trained both now and in the future for the examples shown; the aim is to demonstrate how the presented method utilises the forward model to improve the estimation of the nonlinear coherence, and so it is not benchmarked against the forward prediction in the traditional sense and the selection of a forward model is somewhat arbitrary.

In the presence of noise, even with a perfect model it is difficult to determine whether the entire response has been captured. The lower bound on the nonlinear coherence is $\mathrm{Co}(Y_z, Y_n)$ and the upper bound is 1. The true nonlinear coherence could lie anywhere within this range, and the results in this section demonstrate that the method presented in this paper provides a relatively accurate estimation that could be useful in practice.

### 4.1    POLYNOMIAL STIFFNESS

Systems with quadratic and cubic nonlinearities appear in many mechanical (Nayfeh & Mook, 1979; Worden & Tomlinson, 2001) and biological systems (Miller, 2005), and $y^2$ and $y^3$ are the next terms in the Taylor expansion of the nonlinearity of any system. This example therefore has wide ranging real world applications. The following dimensionless nonlinear oscillator was simulated to generate 1000 frames of length 6000:

$$\ddot{y} + 2\zeta\dot{y} + y + \alpha_2 y^2 + \alpha_3 y^3 = x \ . \tag{24}$$

The input, $x$, was bandlimited random noise with a fixed rms $\tau$. The spectrum of $x$ was effectively flat in the range of interest. The hyperparameters were set as defined in Appendix C.1.

The linearised response of the system captured 88 % of the total system response and a CNN (kernel width=20, hidden layers = 5, features=10, trained for 1000 epochs with a learning rate of 0.01) captured $94\%$ of the response using 10 frames of data. Without the method presented in this paper, with just 10 frames of data it would be impossible to determine how much of the unmodelled response is due to noise and how much is due to nonlinearities. Figure 2 shows the true nonlinear coherence (solid black line), the prediction of the CNN in the forward direction (blue dotted line) and the linear coherence between $x$ and $y_n$ (solid green line). The more noise that was added, the lower the true nonlinear coherence because the causal relationship between $x$ and $y_n$ is weaker. The predicted nonlinear coherence was calculated using 10 frames of data, which is represented by the red dashed line. The method presented in this paper provides an excellent prediction of the nonlinear coherence. For this example system, the forward prediction captures a large component of the response; the method confirms this as in each case the predicted coherence is only slightly closer to 1 than that of the forward model. This is especially important in the high noise example, as if the forward model was trained on this noisy data, the loss would appear large and researchers may be lead to believe that they should collect more data and build larger models to improve the

performance. However, the estimation provided by the presented method correctly indicates that most of the error is due to additive noise and only relatively small improvements to the model are possible.

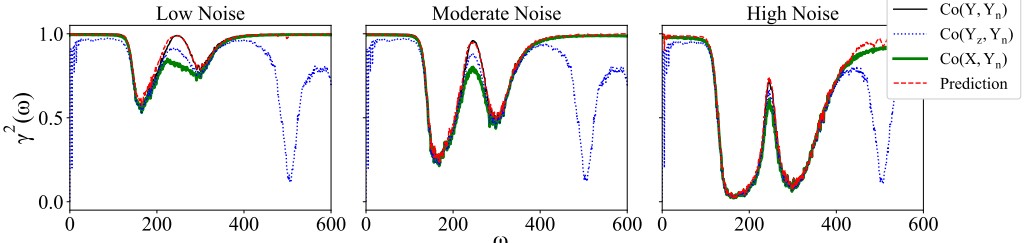

Figure 2: A comparison between the true nonlinear coherence (black solid) and the prediction (red dashed), for the polynomial stiffness case study, with three levels of noise.

## 4.2 SATURATING STIFFNESS

Saturating nonlinearities are common in hardware applications, such as Opamps, transistors, springs, and motors (Mellodge, 2016; Liu & Michel, 1994). The following dimensionless nonlinear oscillator containing a saturating nonlinear stiffness term was used to generate 1000 frames of length 6000:

$$\ddot{y} + 2\zeta\dot{y} + \alpha_1 y + \alpha_2 \tanh\left(10^4 y\right) = x \,. \tag{25}$$

The input, $x$, was bandlimited noise with an rms of $\tau$. The spectrum of $x$ was effectively flat in the range of interest. The hyperparameters are set as in Appendix C.2.

The linearised response of the system captured 58 % of the total system response and a CNN captured 65 % of the response using 10 frames of data (kernel width=10, hidden layers = 5, features=3, trained for 100 epochs at a learning rate of 0.01), hence this is a strongly nonlinear system. Figures 3 shows the nonlinear coherence estimation calculated using 10 frames of data and

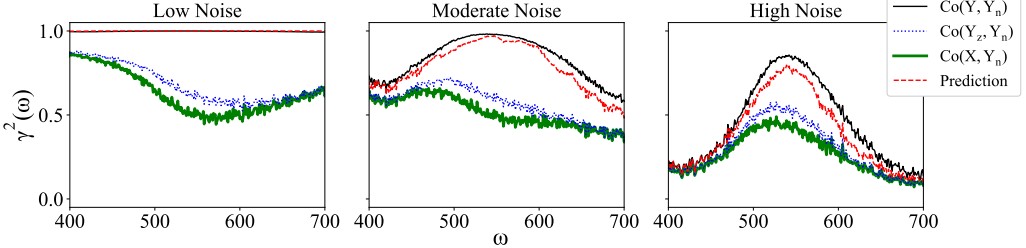

Figure 3: A comparison between the true nonlinear coherence (black solid) and the prediction (red dashed), for the saturating stiffness case study, with three levels of noise.

the forward prediction. In this case the forward model does not capture a significant component of the response, which is indicated by the method. For all three levels of noise, the presented method provides a significant improvement over the lower bound Co $(Y_z, Y_n)$ and the upper bound of 1. The nonlinear coherence is underestimated for the high noise example because, in this case, the estimation of $\hat{y}$ that can be calculated using $y_n$ and $y_z$ is relatively poor. This biases the mapping learnt from $y$ to $x$. The estimation of the nonlinear coherence would be improved by building a more accurate forward model, but this would require more resources. However, the estimation of the nonlinear coherence, even in the high noise case, is still relatively good and the general shape of the curve is captured.

## 4.3 NONLINEAR FRICTION

Friction is ubiquitous across engineering, but can be strongly nonlinear. The following dimensionless nonlinear oscillator containing a Coulomb friction term (Ellis, 2012) was used to generate 1000

frames of length 6000:

$$\ddot{y} + \alpha_1 y + \alpha_2 \tanh\left(10^4 \dot{y}\right) = x \ . \tag{26}$$

The input, $x$, was bandlimited noise with an rms of $\tau$. The spectrum of $x$ was effectively flat in the range of interest. The hyperparameters are set as in the Appendix C.3.

The linearised response of the system captured 91 % of the total system response and a CNN captured 88 % of the response using 10 frames of data (kernel width=10, hidden layers = 5, features=3, trained for 1000 epochs at a learning rate of 0.01), hence this is a moderately nonlinear system. Figures 4 shows the nonlinear coherence estimation calculated using 10 frames of data and

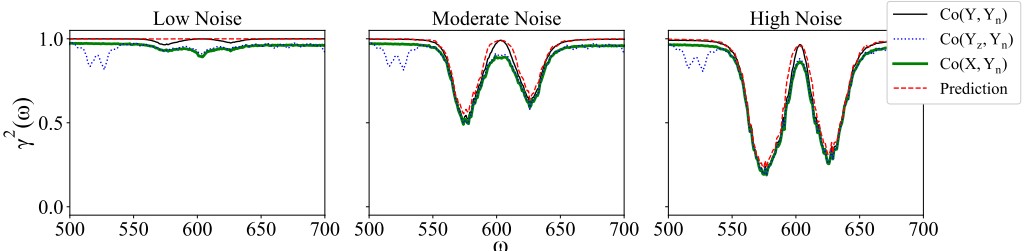

Figure 4: A comparison between the true nonlinear coherence (black solid) and the prediction (red dashed), for the nonlinear friction case study, with three levels of noise.

the forward prediction. In this case, the forward model captures most of the system response and so only small improvements to the model are possible. The presented method demonstrates this and provides an excellent estimation of the nonlinear coherence for each noise case.

## 5 EXPERIMENT

The method was then evaluated on an experimental dataset consisting of 360 frames of length 6000. This demonstrates the applicability of the method to an arbitrary nonlinearity. A cantilever was excited through a nonlinear connection using a bandlimited random input. The nonlinear connection consisted of magnets (the cantilever is steel) and a rubber tip which caused rattling. This nonlinear connection is complex and has not been characterised. A single mode of the cantilever was excited, so it effectively had one degree-of-freedom. An accelerometer was placed on the shaker as a reference signal, $x$, and a second accelerometer was placed along the cantilever as a target signal, $y_n$. The set up was considered to be effectively noiseless and then three levels of additional noise were introduced using post-processing to represent uncorrelated noise in the system. Noise was introduced artificially because for this experimental rig it is difficult to introduce mechanical noise that is uncorrelated with the input due to strong coupling between inputs. Two minutes of data (10 frames) was used to generate the forward prediction and then predict the nonlinear coherence. See Figure 10 in Appendix C.4 for the full experimental setup.

Given the nonlinearity in the system and small quantity of data captured, in the noise-free case only 45 % of the response was captured by the linearised component of the response, and a CNN (kernel width=10, hidden layers = 5, features=5) captured just 74 % due to the complexity of the system response. When noise is added, the challenge is to determine the component of $y_n$ that is caused by $x$, as this represents the component of $y_n$ that could theoretically be reduced using an ANC system. Figure 5 shows the predicted and true nonlinear coherence for three levels of noise. For all three levels of noise, the predicted nonlinear coherence was an excellent estimation of the true nonlinear coherence. In the low noise case, it demonstrates that significant improvements in the prediction could be achieved by collecting more resources and building larger models. Conversely, in the high noise case, the potential improvements are relatively small compared with the overall noise level. This demonstrates the insight provided by the presented method; in an ANR application, it may be worth investing resources in the low noise case to develop a new product, but not in the high noise case.

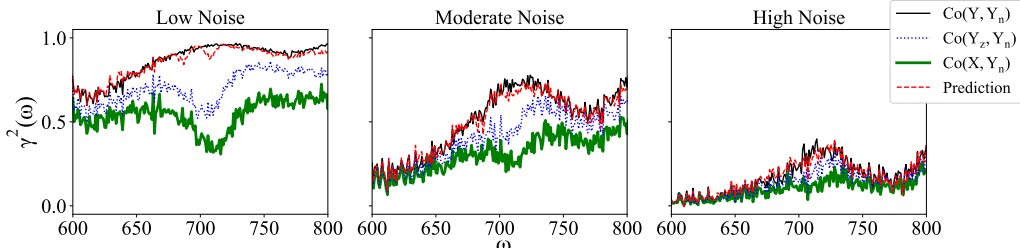

Figure 5: A comparison between the true nonlinear coherence (black solid) and the prediction (red dashed), for the experimental case study, with three levels of noise.

## 6 LIMITATIONS

The main challenge in the work is determining the optimal value for $\lambda$. The intuition is that $\mathcal{L}_x^V$ increases slowly as $\lambda$ increases until the nonlinear coherence prediction is close to the true nonlinear coherence, and then it begins to increase more rapidly. This can be seen in Appendix D. However, it is difficult to determine this transition point and accurately identify the optimal value for $\lambda$. Across the range of examples tested, thresholding $\mathcal{L}_x^V$ worked well empirically and the method was robust to small changes in the threshold value, but an interpretable method for setting $\lambda$ would improve this work.

Furthermore, another limitation relates to the quality of the forward model prediction and the level of additive noise. Define $\delta$ such that $\hat{Y} = Y + \delta$ and then consider:

$$J = E\left[|\hat{Y} - Y|^2\right] = E\left[\delta\delta^*\right] \tag{27}$$

and

$$\mathcal{L}_x^T = \sum_{i=1}^{N^T} \frac{||\mathcal{F}_\theta\left(\hat{y}^{(i)}\right) - x^{(i)}||^2}{\sigma_x^2} = \sum_{i=1}^{N^T} \frac{||\mathcal{F}_\theta\left(y^{(i)} + \delta^{(i)}\right) - x^{(i)}||^2}{\sigma_x^2} \ . \tag{28}$$

The key assumption of the method is that minimising $\mathcal{L}_x^T$ is equivalent to minimising $J$, which depends on two factors. Firstly, in order to minimise $\mathcal{L}_x^T$, $\delta$ must be such that the argument of $\mathcal{F}_\theta$ is close to $y$. If $\delta$ is very large, this causes bias in the trained $\mathcal{F}_\theta$ which reduces performance. This happens if both the model error and the level of additive noise are very high. Secondly, both $y_z$ and $y_n$ must contain useful information to infer $x$, otherwise the method cannot provide a useful estimation of the nonlinear coherence: specifically $y_z$ must at least include a good estimate of the linear component of $y$, which is not difficult to achieve. This is because it needs to be driven to incorporate both signals.

## 7 CONCLUSIONS

This paper presents a novel method for calculating the nonlinear coherence for a nonlinear dynamical system in the presence of noise. A set of parameters are learnt to optimally combine an output prediction, calculated using an available model, with noisy measurements of the output to predict the input to the system. The nonlinear coherence is calculated using these parameters and is used as a metric of causality. Previously, the only way to estimate this was to model the system and assume the unmodelled component of $y_n$ to be noise. For highly nonlinear systems, this can be a poor estimation of the nonlinear coherence. The presented method improves upon this and is able to estimate the nonlinear coherence with excellent accuracy using a relatively small quantity of data. This would allow for users to collect experimental input and output data for a particular nonlinear dynamical system, and identify how much of the output is caused by the input across the frequency range, without developing expensive models. The method works for a broad class of nonlinear dynamical systems which are widely applicable across many scientific disciplines, from structural dynamics to neuroscience. The main limitation of the method is the interpretability and selection of the hyperparameter $\lambda$. Future work should focus on this and investigating the applicability of the method to a wider class of nonlinear dynamical systems.

REPRODUCIBLITY STATEMENT

The code and datasets required to reproduce the results in this paper are shared at https://github.com/obj22/repo-1. The hyperparameter settings for the simulations are provided in the appendix.

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

## A    CONTROLLED INPUTS

If the inputs to a system can be controlled, the power spectrum density of the deterministic component of the response, $y$, can be calculated. Consider two noisy observations of $y(t)$, caused by the same input $x(t)$:

$$y_n(t) = y(t) + \epsilon_n \qquad\qquad y_m(t) = y(t) + \epsilon_m \ . \qquad (29)$$

In the frequency domain, the power spectrum density of the deterministic component is defined to be

$$S_{yy}(f) = E\left[YY^*\right] \qquad (30)$$

Consider the cross-spectrum between $y_n$ and $y_m$:

$$S_{y_n y_m}(f) = E\left[Y_n Y_m^*\right] = E\left[YY^*\right] + E\left[\mathcal{E}_n Y_m^*\right] + E\left[Y_n \mathcal{E}_m^*\right] + E\left[\mathcal{E}_n \mathcal{E}_m^*\right] \qquad (31)$$

$$= E\left[YY^*\right] \ . \qquad (32)$$

Therefore by taking the cross-spectrum between the two measured outputs, the deterministic component of the response can be identified. This is not applicable in this paper because repeat measurements are not available and $\mathcal{E}_z$, as defined in Section 3.2, is correlated with Y.

## B    ARCHITECTURE

### B.1    DERIVATION

The architecture is driven towards de-noising $\hat{Y}$ by optimally combining $Y_n$ and $Y_z$.

$$J = E\left[(\hat{Y} - Y)(\hat{Y} - Y)^*\right] \qquad (33)$$

$$= E\left[(Y_n K + Y_z(1-K) - Y)(Y_n K + Y_z(1-K) - Y)^*\right] \qquad (34)$$

$$= K^2 E\left[\mathcal{E}_n \mathcal{E}_n^*\right] + (1-K)^2 E\left[\mathcal{E}_z \mathcal{E}_z^*\right] \ . \qquad (35)$$

Then, differentiating with respect to $K$ gives

$$\frac{dJ}{dK} = 2K E\left[\mathcal{E}_n \mathcal{E}_n^*\right] - 2(1-K) E\left[\mathcal{E}_z \mathcal{E}_z^*\right] = 0 \ . \qquad (36)$$

Therefore rearranging for $K$ gives

$$K = \frac{1}{1 + \frac{E[\mathcal{E}_n \mathcal{E}_n^*]}{E[\mathcal{E}_z \mathcal{E}_z^*]}} \ . \qquad (37)$$

## C    DATASET

The dataset is split into training, validation and test sets. 10 frames were used for training, 10 frames were used for validation, and remaining frames were used for the test set. A batch size of 1 frame was used for training using the Adam optimiser. For each simulation, fourth order Runge-Kutta numerical integration was used in order to calculate the benchmark true $y$ using $x$ (Press et al., 2007).

### C.1    POLYNOMIAL STIFFNESS CASE STUDY 1

The dataset contains 1000 frames of length 6000 samples. The parameters were set as: $\zeta = 4.5$, $\alpha_1 = 5 \times 10^3$, $\alpha_2 = -10$ and $\alpha_3 = 3 \times 10^3$. The input, $x$, was bandlimited noise with an rms of $\tau = 1.0 \times 10^3$. A step of 0.0025 was used in the numerical integration. Figure 6 shows the power spectrum density (PSD) for the system response and the linearised response.

### C.2    SATURATING STIFFNESS CASE STUDY

The dataset contains 1000 frames of length 6000 samples. The parameters were set as: $\zeta = 3.0$, $\alpha_1 = 10^4$ and $\alpha_2 = 5 \times 10^3$. The input, $x$, was bandlimited noise with an rms of $\tau = 6.0 \times 10^3$. A step of 0.005 was used in the numerical integration. Figure 7 shows the power spectrum density (PSD) for the system response and the linearised response.

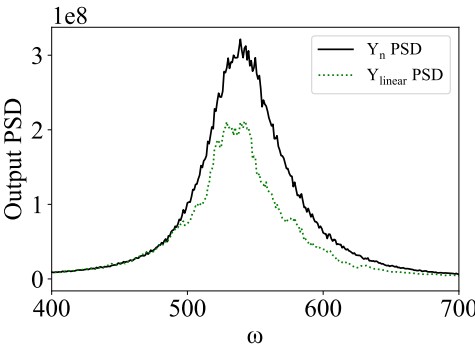

Figure 6: Plot of the power spectrum density of $Y_n$ and $Y_z$ for the polynomial stiffness case study with no noise added.

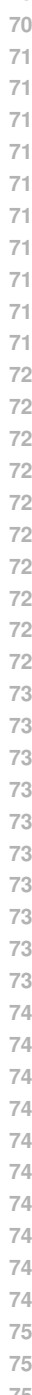

Figure 7: Plot of the power spectrum density of $Y_n$ and $Y_{linear}$ for the nonlinear damping data with no noise added.

### C.3 NONLINEAR FRICTION CASE STUDY

The dataset contains 1000 frames of length 6000 samples. The parameters were set as: $\alpha_1 = 10^5$ and $\alpha_2 = 0.5$. The input, $x$, was bandlimited noise with an rms of $\tau = 5$. A step of 0.002 was used in the numerical integration. Figure 8 shows the power spectrum density (PSD) for the system response and the linearised response.

### C.4 EXPERIMENT CASE STUDY

The dataset contains 180 frames of data of length 6000. A sampling frequency of 500 Hz was used to record the data. Figure 9 shows the power spectrum density (PSD) for the system response and the linearised response. The full experimental set up is shown in Fig. 10.

## D $\lambda$ PLOTS

Figures 11, 12, 13 and 14 show plots of how $\mathcal{L}_x^V$ varies as $\lambda$ increases. Plotted against the ratio of the two coefficients in the loss function, $\frac{\lambda}{1-\lambda}$, because it is the relative weighting between the two terms that is important. The red circle in each plot indicates the point at which $\mathcal{L}_x^V$ crosses the threshold of 0.001 above the minimum value. This is the value of $\lambda$ used to calculate the nonlinear coherence.

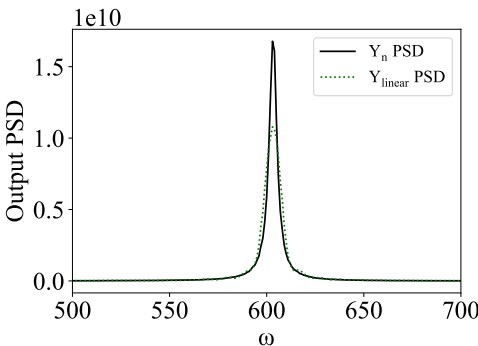

Figure 8: Plot of the power spectrum density of $Y_n$ and $Y_{linear}$ for the nonlinear friction data with no noise added.

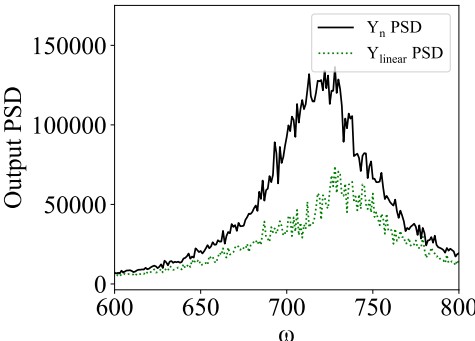

Figure 9: Plot of the power spectrum density of $Y_n$ and $Y_{linear}$ for the experimental data with no noise added.

# E    COMPUTATIONAL RESOURCES

The device used to run the code in this paper had the following specifications:

Processor: 12th Gen Intel(R) Core(TM) i7-12700KF 3.61 GHz

RAM: 64.0 GB

GPU: NVIDIA GeForce RTX 3090

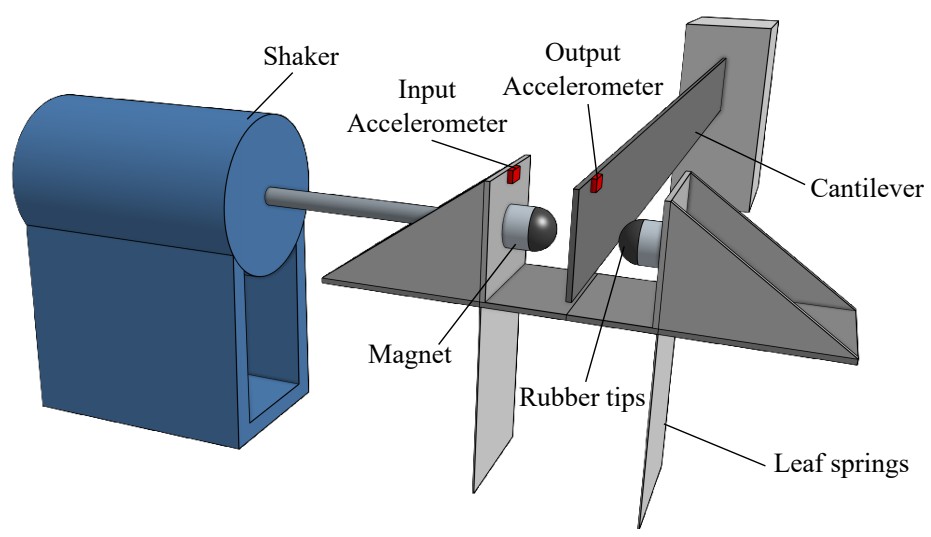

Figure 10: Diagram of the experimental set up.

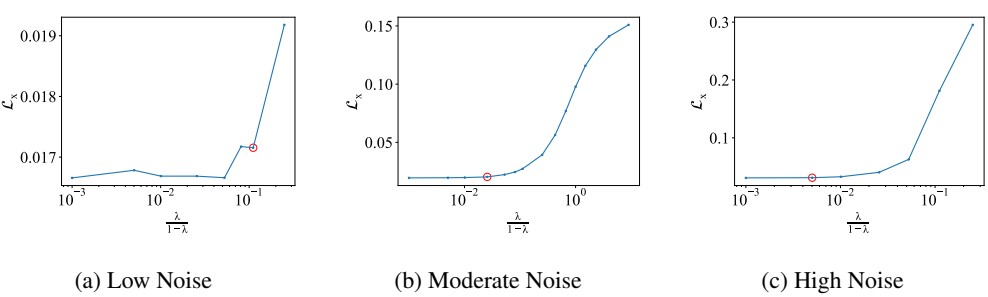

(a) Low Noise                    (b) Moderate Noise                    (c) High Noise

Figure 11: Polynomial stiffness case study 1.

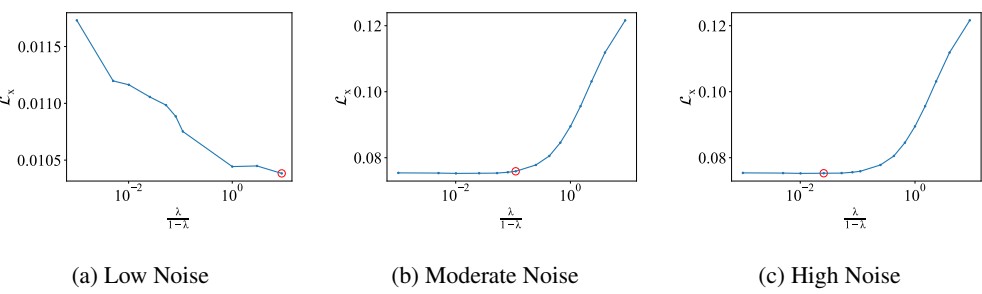

(a) Low Noise                    (b) Moderate Noise                    (c) High Noise

Figure 12: Saturating stiffness case study.

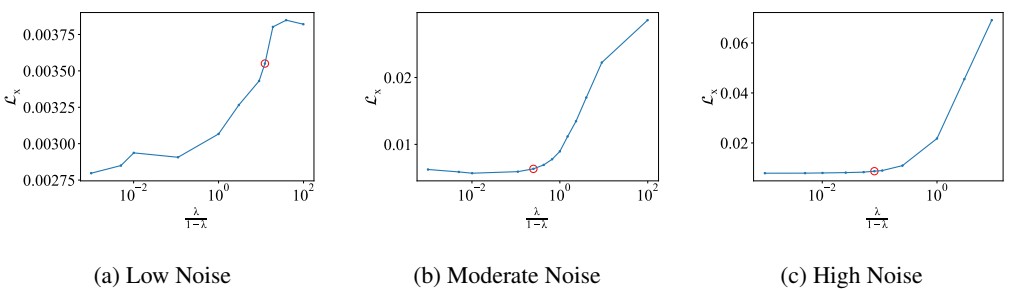

(a) Low Noise        (b) Moderate Noise        (c) High Noise

Figure 13: Nonlinear friction case study.

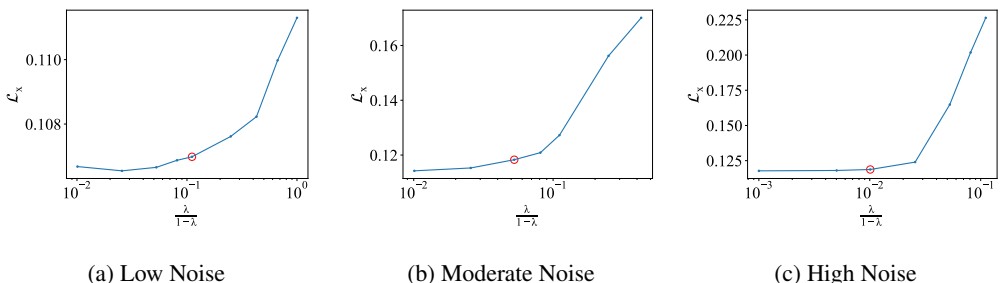

(a) Low Noise        (b) Moderate Noise        (c) High Noise

Figure 14: Experimental case study.