# OpenReview forum: "A method for identifying causality in the response of nonlinear dynamical systems"
_ICLR.cc/2025/Conference — Submitted to ICLR 2025_

### Official Review · Reviewer_Wwmi · 2024-10-25

**Soundness:** 2
**Presentation:** 1
**Contribution:** 2
**Rating:** 3
**Confidence:** 2

**Summary:**

The authors of the manuscript ‘A METHOD FOR IDENTIFYING CAUSALITY IN THE RESPONSE OF NONLINEAR DYNAMICAL SYSTEMS’ develop a method for the calculation of the nonlinear coherence between input and output data for a noisy ODE class given random input. The approach allows to learn an optimal combination of an output prediction with noisy measurements. The nonlinear coherence that is calculated is then used as a measure of causality. The method is shown to perform well on both simulated dynamical datasets as well as experimental data.

**Strengths:**

- The approach is well characterized mathematically in the manuscript.
- The demonstration of the approach over different types of simulated and experimental data is interesting.
- There are comparisons to alternative approaches over the same task.

**Weaknesses:**

- The manuscript could be improved in terms of readability. It is in many cases very difficult to follow.
- While the method was evaluated over real experimental data, the noise was added synthetically.

**Questions:**

Comments:

- The description of the problem and approach in the abstract is not clear enough.
- Could you provide a more thorough statistical analysis of the performance of the approach for the simulated datasets over different parameter regimes of the underlying model, to show its robustness?
- The figures are missing captions.
- It would be informative, for the simulated and experimental results, to explicitly show how a better approximation of the response (as presented in the figures) helps in the interpretation of the original systems.

Minor comments:
- ‘but has recently begun’ - the ‘but’ part of the sentence should probably be rephrased
- it should be mentioned in the abstract that this work is restricted to a specific class of ODEs.
- ‘A completely new strategy’ - rephrase?
- The related work section sounds too defensive.. e.g. ‘The presented method .. is very different to work in the literature.’
- ‘In this paper, the primary aim is to estimate the noise level and the model error, but a similar principle is utilised.’ - unclear.
- ‘The presented method.. provides an excellent estimation of the nonlinear coherence for each noise case’ - rephrase in a more quantitative way?

---

> ### Author Response · Authors · 2024-11-23
> **Response to Reviewer Wwmi 1.0**
>
> We thank the reviewer for their helpful, constructive comments. We respond to your individual points below.
>
> Weaknesses:
>
> 1) The reviewer notes that “the manuscript could be improved in terms of readability. It is in many cases very difficult to follow.” The updated manuscript has been updated throughout to improve readability.
>
> 2) The reviewer identifies the use of synthetic noise in the experimental dataset as weakness. This was unavoidable for our experimental set up due to coupling between the potential noise generating processes and the input shaker. The aim of the experimental data was to demonstrate the performance of the method on an arbitrary nonlinearity, rather than something defined in a simulation. Therefore we do not believe the source of the noise is critical to the value that this benchmark provides, though note that this would improve future experiments and will be considered for future work.
>
> Questions:
>
> 1) The reviewer comments on the lack of clarity in the abstract. In the updated manuscript the abstract has been significantly changed to improve readability.
>
> 2) The reviewer asks whether it is possible to “provide a more thorough statistical analysis of the performance of the approach for the simulated datasets over different parameter regimes of the underlying model, to show its robustness.” We have chosen to demonstrate its robustness by showing its performance over a range of nonlinear dynamical systems and noise levels. Aside from having different functional forms of nonlinearities, these systems have different levels of nonlinearity relative to each other. This was preferred over choosing one system and varying its parameters to adjust the level of nonlinearity, and it is not feasible to do both within the page limit.
>
> 3)	The reviewer notes that “the figures are missing captions.” We accept that the captions are brief  and have extended them to improve the readability of the paper.
>
> 4)	The reviewer suggests that “it would be informative, for the simulated and experimental results, to explicitly show how a better approximation of the response (as presented in the figures) helps in the interpretation of the original systems.” To address this, the section on ANR has been expanded (lines 66-78) to show how the nonlinear coherence links to the performance of an ANR system. There is insufficient space to include figures of this in the manuscript, but hopefully this is clearer in the updated manuscript.
>
> We thank the reviewer for the additional, minor comments provided and have updated the manuscript accordingly in each case.

---

> > ### Comment · Reviewer_Wwmi · 2024-11-27
> >
> > I appreciate the revisions made by the authors to address comments and suggestions.
> >
> > I would like to point out that the fourth question, "It would be informative, for the simulated and experimental results, to explicitly show how a better approximation of the response (as presented in the figures) helps in the interpretation of the original systems" is not addressed by an extended introduction paragraph because it refers to the results themselves.

---

### Official Review · Reviewer_1Yvb · 2024-11-04

**Soundness:** 2
**Presentation:** 2
**Contribution:** 2
**Rating:** 3
**Confidence:** 2

**Summary:**

The manuscript presents a novel technique for separate measurement noise and modeling errors without accessing the actual input-output relationship. The calculation was performed in frequency domain and there's one weighting parameter per frequency controlling the contribution of modeling error and measurement error to achieve minimum reconstruction error. The performance, measured by the coherence between actual signal and recovered signal is high.

**Strengths:**

- The presented method learns the a nonlinear functional mapping through a neural network. It works under a wide range of nonlinearities form as long as y can be mapped back to x (Eq.3).
- It presented a novel way to balance modeling / observation error and contributed to maximize signal reconstruction performance in highly nonlinear systems.
- Even in high noise cases, the recovered coherence was still high.

**Weaknesses:**

- Only applies to the specific ODE described in Eq.3. Maybe the authors could extend it further as long as there's a one-to-one mapping from y to x?
- the method only works from 1D signal to 1D signal? Is there any possibility to extend to high dimensional nonlinear systems.

**Questions:**

- What is the relationship between K and \lambda? Is K a hyper parameter during network training? How did you find the optimal K.
- Why the modeling error / measurement error affects the signal in an additive way (Eq.4)

---

> ### Author Response · Authors · 2024-11-23
> **Response to Reviewer 1Yvb 1.0**
>
> We thank the reviewer for their helpful, constructive comments. We respond to your individual points below.
>
> Weaknesses:
>
> 1)	The reviewer states asks if the work can be extended “further as long as there's a one-to-one mapping from y to x” and if it applies to systems with multiple degrees-of-freedom. This is correct. The method works provided there is a simple mapping from y to x, with no nonlinear memory terms. In addition, the method works for systems with multiple degrees-of-freedom, provided measurements are taken across the nonlinearity in the system. This prevents the memory of the reverse mapping being long and nonlinear. This is noted and explained in the updated manuscript in lines 81-88 and 171-175.
>
> Questions:
>
> 1)	The reviewer asks what the relationship is between $\lambda$ and K, and asks how K is calculated. K is trained alongside the network weights. This is clarified with additions to the manuscript in lines 191-192. $\lambda$ is used as part of the training process in order to bias K towards 1, as discussed in section 3.4. Its value is chosen by considering how the validation loss changes as $\lambda$ increases. K is trained, along with the CNN for each value of $\lambda$, and then for the selected $\lambda$, the corresponding K is used to calculate the nonlinear coherence. See the additions in line 310.
>
> 2)	The reviewer asks “why the modelling error / measurement error affects the signal in an additive way (Eq.4)”. The assumption in this work is that $\epsilon_n$ is due to uncoupled, unmeasured noise sources that additively contribute to the total output. See lines 43-44 for additions.

---

### Official Review · Reviewer_3SzE · 2024-11-04

**Soundness:** 3
**Presentation:** 1
**Contribution:** 2
**Rating:** 5
**Confidence:** 2

**Summary:**

The paper introduces a method for estimating the coherence $Co(y, y_n)$ between the observed noisy data $y_n$, and the “true” underlying system $y$. This estimate, termed $\hat{y}$ is obtained by combining, in frequency domain, a model/input based estimate $y_z$, with the noisy data $y_n$ (similar to a Kalman filter). The estimated $Co(y, y_n)$ is closer to $Co(y, y_n)$, than an only model/input-based estimate $Co(y_z, y_n)$.

**Strengths:**

- The method uses a sensible way to combine model/input based data with observed data.

- The resulting method outperforms the chosen baselines in terms of the coherence metric.

**Weaknesses:**

1. I found the paper hard to read, I think this can be improved by the authors more clearly explaining or motivating their motivation for the different steps (which might be obvious to them, but might not be to the reader), e.g.,
    - 1.1 “Causality” is in the title and a key point of the paper, however I am totally missing some explicit motivation for why measuring coherence is the best way to estimate causality. The one citation for coherency (Silva et al., 2016), seems an odd choice.
    - 1.2 ANR introduced, but it, and the cited papers seem only tangentially related. This could be scrapped to make space for motivating the actual method, i.e., the use of coherency.
     - 1.3 No motivation is given for combining a model-based estimate with data in the frequency domain.
     - 1.4 Altough $\mathcal{F}$ of Eq. 1 is called a function, it is not a map from $x(t)$ to $y(t)$ (then there would be no dynamical system). Relatedly, it might also be better to use a different characters for $\mathcal{F}$ and $\mathcal{F}_\theta$, which are two very different things.

2. I find it very confusing that the paper introduces, and keeps referring to non-linear coherence, when the paper really seems to be after estimating the “standard”, linear coherence between $y$ and $y_n$.

3. Page 9, has half its space dedicated to a Figure of an experimental setup, but this is just another benchmark, I would suggest to make this Figure smaller (or move it to appendix), and use the remaining space to improve readability.

4. The only evaluation here is $Co(\hat{y},y)$, against other coherences, which is not immediately intuitive. Could the authors not simply also plot and/or quantify how close $\hat{y}$ is to $y$?

5. Fig. 1 seems to lack a $\hat{Y}$ before, and $\hat{y}$ after the IFFT box. $X$ and a box with $\mathcal{F}$ (with some parameters $\theta$?) feeding into $y_z$ would also be helpful.

6. I find the motivation for needing $\mathcal{L}y$ and $\lambda$ in general a bit lacking in theoretical foundations. A possible perspective could potentially be that Eq. 5 gives one the variational estimate of the true distribution of $y$ as $q(\hat{y}|y_n, x)$,  $F_\theta$ gives one $p(x|\hat{y})$ as $\mathcal{N}(x; F_\theta(\hat{y}), \sigma_x^2)$. If one adds a prior on $\hat{y}$ as p$(\hat{y})=\mathcal{N}(y_n, \sigma_y^2)$ and writes out the typical ELBO loss on $p(x)$, both terms in Eq. 7 drop out (just the entropy of $q$ is missing) - although I am sure other perspectives are possible!

**Questions:**

1. Why is Wavenet a good baseline? If one wants to distinguish what combining the forward model with a data inferred state brings you, can one not simply take the forward model from section 3.3?
2. Are the $\dot{x}$ and $\ddot{x}$ terms in Eq. 3 needed (they seem to be absent in all benchmarks)?
3. Eq. 7, is there an implicit Gaussian assumption?
4. Line 197 Why would $\lambda=0$ yield the optimal $K$, this is not obvious to me?

---

> ### Author Response · Authors · 2024-11-23
> **Response to Reviewer 3SzE 1.0**
>
> We thank the reviewer for their helpful, constructive comments. We respond to your individual points below.
>
> 1) We note that the reviewer found the paper hard to read and suggested improving the description of the motivation and method. To address this, significant changes have been made to the introduction. Individual points within the introduction are addressed below.
>
> 1.1) Linear coherence measures the component of the signal $y_n$ which can be predicted using the signal $x$. In this context, given $x$ is the input to the system and if the system is assumed to be linear, the linear coherence provides a measure, between 0 and 1 of how much of $y_n$ is caused by $x$. Computing the linear coherence relies on a linear model of the system being simple to compute; it is assumed that the reduction of the linear coherence below 1 is caused by noise, because the linearisation can be computed with a relatively small quantity of data. The nonlinear coherence is therefore more difficult to compute because building a nonlinear model of the system is more complex, and so differentiating between the unmodelled nonlinearity and the additive noise is challenging. Currently, a highly accurate model must be built to eliminate model error in order to estimate the level of additive noise. Our method provides a way to estimate this with a small quantity of data and without a high accuracy model. The nonlinear coherence provides a method to estimate how much of the output can be predicted from the input, and so can guide investment of resources into improving models; if the nonlinear coherence is low, then it may not be worth collecting more data because the best possible may not provide sufficient performance for the application. In addition, we note the suggestion for the citation and have selected a better one.
>
> We realise the use of the word `causality’ is strong; here we use it to mean how much of the output is predictable using the input. In this case, if x is the input to the system, governed by Eq. (3), then x does cause y. In general, the method provides a measure of how much of the output can be predicted using the input and the level of noise. See the changes to the lines 53-65.
>
> 1.2) The reviewers stated that they believe the section on the application of the work to ANR is only tangentially related. This section was included because ANR provides an application of the method in a commercial setting and demonstrates the usefulness of the method. We believe it is also a useful application to explain the problem set up. For a linear system, if the linear coherence between the reference signal and the target signal is low, this indicates that a high level of noise reduction is not possible. There is a direct relationship between the linear coherence and the level of noise reduction possible:
> $$\Gamma_{XY}(f)=10log_{10}{\left(1-\gamma_{XY}^{2}\right)} \ ,$$
> where $\Gamma_{XY}(f)$ is measured in decibels. For a nonlinear system, it is not currently possible to compute the nonlinear coherence; more resources must be invested to build more accurate models to explore whether improvements are possible. Our method provides an estimation of the improvements that could be possible with more data, without having to collect more data. We have therefore left this section in the paper but have expanded it to explain its significance in more detail. We have also reworded the section to improve readability and emphasise the significance of the application. Please see lines 65 to 78.
>
> 1.3) The reviewer states that the motivation for combining a model-based estimate with data in the frequency domain is not clear. The motivation for this is based on the approach used in Kalman filters. The filter optimises state estimation by balancing the prediction of a model and measured signal using their relative error levels. If the quality of state estimation can be evaluated, then the level of noise and model error can be identified. However, typically this is not possible if only $y_n$ is available, as we cannot evaluate the estimation of y because it is not available. However, in our case, x can be used as a reference signal for this process. Therefore, the model based estimate and measured data can be combined, as in the Kalman filter, and the estimation of y is evaluated via the mapping to x.  This has been added to the manuscript in lines 183-188.
>
> 1.4) We accept this point and have adjusted the notation and language accordingly.

---

> ### Author Response · Authors · 2024-11-23
> **Response to Reviewer 3SzE 1.1**
>
> 2) The reviewer states that the importance of the nonlinear coherence is not clear and that the linear coherence between $y$ and $y_n$ is presented as the important quantity. The linear coherence between $y$ and $y_n$ does represent the nonlinear coherence between $x$ and $y_n$, because it is the component of $y_n$ that can in principle be predicted using $x$. The problem is that y is not available. The coherence $\gamma^2_{YY_n}(f)$ is therefore a convenient term that can be estimated using our method, and can be interpreted as the nonlinear coherence. This has been explained more clearly in lines 53-65.
>
> 3) We accept that the experimental figure is not essential and have moved the figure to the appendix to make space to improve readability in other sections.
>
> 4) The reviewer suggests that $\text{Co}(\hat{Y},Y_n)(f)$ is not intuitive as a metric and that it may be more appropriate to quantify how close $\hat{y}$ is to $y$. However, $\text{Co}(\hat{Y},Y_n)(f)$ is not the metric. The prediction of the nonlinear coherence, i.e. $\text{Co}(Y,Y_n)(f)$, is the key metric, which calculated using the estimated noise levels. The estimated value of Y is not useful in itself and so comparing it to Y does not provide much information. The algorithm is not trying to provide a better estimation of Y, but an estimation of how much of $Y_n$ it is possible to estimate in theory, using $X$. This is quantified by $\text{Co}(Y,Y_n)(f)$, as a value between 0 and 1, where 0 represents that none of $Y_n$ can be predicted using $x$ at that frequency, and 1 indicates it can be fully predicted. Additions in the lines 53-65 aim to improve the interpretability of $\text{Co}(Y,Y_n)$ and nonlinear coherence. Additions in lines 334-335 clarify that the method is predicting $\text{Co}(Y,Y_n)(f)$ rather than $\text{Co}(\hat{Y},Y_n)(f)$.
>
> 5) The reviews suggest some improvements to Figure 1. We accept these suggestions and have adjusted the figure accordingly.
> The review states that the motivation for $\mathcal{L}_{y}$ and $λ$ is lacking and suggests a variational estimate could be a possible perspective. We have considered this perspective, but there do not believe it is valid in our case. Firstly, we assume the input is noiseless because the source of noise relevant to many applications is outputs caused  by unmeasured inputs. As a result, in the equations presented by the reviewer, $\sigma_x^2=0$. The error in the prediction is due to the reverse model mismatch because of the error in y.  It is not clear that this translates to the parameter , and so we have chosen the approach presented, as a pragmatic and empirical solution that provides useful insight.
>
> Questions:
> 1) The reviewer asks why wavenet is a good baseline and suggests taking the forward model defined in section 3.3 as the benchmark. This is the approach, and additions to lines 337-341 aim to clarify this.
>
> 2) The reviewer asks whether $\dot{x}$ and $\ddot{x}$ are needed in Eq. (3) because they don’t appear in the benchmarks. The method generalises to this case because the introduction of $\dot{x}$ and $\ddot{x}$  only introduces linear memory, which is a simple mapping. However, we accept your point and have removed the terms to avoid confusion.
>
> 3) The reviewers ask if there is an implicit gaussian assumption in Eq. 7. There is no Gaussian assumption, the x and yn signals are random signals with a mean of zero and are normalised to have a variance of 1. This has been clarified in line 201.
>
> 4) The reviewer asks why $λ=0$ results in the optimal value of K. Given the mapping exists between y and x, for $\hat{y} = K y_n + (1 - K) y_z$, the noise on $\hat{y}$ must be reduced as much as possible in order to minimise the error in the prediction of x. This is because noise will only deteriorate the estimation of x. Therefore the optimal K will occur when y is as close to y as possible. Hence $ E\left((y - \hat{y})^2\right)$ is minimised, which leads to the optimal K derived in the paper. See the change to the sentence on line 210 which aims to clarify this point.

---

> > ### Comment · Reviewer_3SzE · 2024-11-25
> >
> > I appreciate the author's willingness to improve the readability and indeed the notation is improved (a latex-diff would be helpful).
> >
> > I do still have have two major concerns / am in need of clarifications.
> >
> > - 2 / 4.  It seems I misunderstood how exactly the model generates the prediction of the "non-linear coherence" (dashed red line in the Figures). Instead of using the CNN's $\hat{y}$, you only use $K(f)$ obtained from the CNN, and plug this in in Eqs. 16,23 and 15? This is an absolutely crucial point, and evidently is not clear at the moment (it would be good to clarify when introducing the simulations).
> > I was previously under the impression that "optimally combining input and observations, meant using $K$ to estimate $y$ as  $\hat{y}$ with $\hat{Y}=KY_n+(1-K)Y_z$, and then using this $\hat{y}$ to estimate $Co(y, y_n)$ as $Co(\hat{y}, y_n)$ which is not what is done? Can the authors confirm that this is indeed different from combining $\hat{Y}_n$ and $Y_z$, with $K$ in Eqs 16-23? A quick demonstration would be to plot $Co(\hat{y}, y_n)$ for one experiment, and show that it is indeed different than your predictions.
> > I find it hard to agree with "The estimated value of $y$ is not useful in itself" - if your method works well, it should also give one the optimal estimate of $y$  as $\hat{y}$?
> >
> > - Q1. These additional lines do little to clarify why the baseline is not exactly the same forward model used in your method. How can we be sure that the improvement of your method comes from the "optimal" combination of input and observations, instead of simply from a better forward model?
> >
> > Finally, regarding point 5 / Q3. I see the that there is no explicit input noise, but in terms of being pragmatic, Eq. 7 is in principle already a log Gauss likelihood assuming input noise. Anyway, this is not the perspective taken in this paper - which is fine - so let's drop this point, and focus on the two critical ones above.

---

> > > ### Author Response · Authors · 2024-11-25
> > > **Response to Reviewer 3SzE 2.0**
> > >
> > > The authors would like to thank the reviewer for taking the time to respond to our rebuttal. We respond to the questions below.
> > >
> > > 1) By combining $Y_n$ and $Y_z$ using $K$, the resulting estimate, $\hat{Y}$, provides a slightly better approximation of $Y$ at each frequency than the least noisy of the two signals. While it is not possible to design a filter that perfectly denoises $Y_n$ and $Y_z$ to recover a clean $Y$, an optimal denoising filter can be constructed to minimize noise as much as possible. The parameters defining this filter are the most valuable result of using our method.
> > >
> > > Even with this optimal filter, $\hat{Y}$ will still contain significant noise. Consequently, while $\hat{Y}$ is the best possible estimate of $Y$, it does not provide much additional insight into the relative noise levels. The key to this method lies in the parameter $K$ which defines the optimal filter, which encapsulates information about the noise levels in each signal.
> > >
> > > Although complete noise removal from $\hat{Y}$ is not achievable, the method provides an estimation of the remaining noise, which is difficult to achieve and provides useful insight.
> > >
> > > Please see the new clarification added to line 334.
> > >
> > > 2) As suggested by the reviewer, the baseline is exactly the same as the forward model. The baseline is $\text{Co}(Y_z,Y_n)$, which uses the forward model, $Y_z$, defined in the method. In this case, $Y_z$ is generated using Wavenet, and this $Y_z$ is used to predict the nonlinear coherence (red dotted line) using the architecture in section 3.1, and it is also used to provide a baseline $\text{Co}(Y_z,Y_n)$ (blue dotted line). The improvement therefore comes from the method. If a better forward model was to be used, the baseline $\text{Co}(Y_z,Y_n)$ would improve, and the estimated nonlinear coherence would also improve.
> > >
> > > Please see the new addition to line 343.

---

> ### Comment · Reviewer_3SzE · 2024-11-25
>
> Thanks for clarifying - these two (crucial) points are now clearer in the manuscript! I will my raise my score slightly.
>
> Indeed $\hat{y}$ is a noisy estimate, but I fail to see why the non-linear coherence that you currently predict is not similarly a noisy estimate. As far as I can tell, both your estimate of $Co(Y,Y_n)$, as well as $Co(\hat{Y},Y_n)$, are calculated from exactly the same quantities ($K$, $Y_n$ and $Y_z$). Could you clarify their (mathematical or empirical) relation?

---

> > ### Author Response · Authors · 2024-11-25
> > **Response to Reviewer 3SzE 3.0**
> >
> > The authors would like to thank the reviewer for their question and increasing their score.
> >
> > Our method gives an analytical derivation of the coherence using the signals provided and the inferred value of K. The suggestion of the reviewer is that $Co(Y,Y_n)$ can be estimated well using $Co(\hat{Y},Y_n)$. If this were true, ideally $Co(Y,Y_n)=Co(\hat{Y},Y_n)$ for the optimal K derived in the paper. However, expanding this equality leads to a very complex expression, and is not easy to simplify this to either prove or disprove the equality. An easier way to provide some intuition as to why the equality is not true is to compare $E((\hat{Y}-Y_n)(\hat{Y}-Y_n)^*)$ with $E((Y-Y_n)(Y-Y_n)^*)$. Firstly:
> > $$
> > E((Y-Y_n)(Y-Y_n)^*)=E(\epsilon_n\epsilon_n^*)
> > $$
> > and then
> > $$
> > E((\hat{Y}-Y_n)(\hat{Y}-Y_n)^*)=E((KY_n+(1-K)Y_z-Y_n)(KY_n+(1-K)Y_z-Y_n)^*)=E(((1-K)(\epsilon_z-\epsilon_n)((1-K)(\epsilon_z-\epsilon_n))^*)
> > $$
> > $$
> > =(1-K)^2(E(\epsilon_z\epsilon_z^*)+E(\epsilon_n\epsilon_n^*))=(1-K)E(\epsilon_n\epsilon_n^*)
> > $$
> > These two expressions are clearly not equal, and so the indication is that the coherences will not be equal. We are not able to upload figures here, but have added figures to show the comparison between the coherence predictions obtained for a couple of systems to the repository in the paper (experimental_figure.png and NLS_1.png). These figures show $\text{Co}(\hat{Y},Y_n)$ overestimates the coherence for the optimized value of K. Even though our estimation is not perfect (the reverse model cannot be perfectly calculated due to noise on $Y_n$), it is analytically correct for the optimal K and the assumption of a perfect model. $\text{Co}(\hat{Y},Y_n)$ is different to this and doesn't have a clear interpretation.

---

> > > ### Comment · Reviewer_3SzE · 2024-11-26
> > >
> > > I want to thank the authors again for their clarification and the additional figures. It is indeed correct that the "optimal" model estimate $\hat{y}$ gives an overestimate of the coherence. Apologies for the additional effort!
> > >
> > > I have to admit that, even after spending considerable time on this manuscript, I am still somewhat confused by the method presented in this paper - where does the quality of the the learned architecture come in, only in how well $K$ is estimated? Does this mean the quality of the dynamics models ($Y_z$) is unimportant? - I fail to see why Eq. 23 does not hold for bad models (e.g., extreme case $y_z \sim \mathcal{N}(0,1)$, independent of $x$)? The paper mention no assumptions except for zero mean error, and no correlation between $e_z$ and $e_n$ and between $e_n$ and $y$.
> > >
> > > My confusion of the methods (even after spending considerable time on this manuscript) can partly be explained by my own failure to grasp this, and I will lower my confidence score. I do want to point out that the readability and interpretability were also major concerns shared by the other reviewer...

---

> > > > ### Author Response · Authors · 2024-11-28
> > > > **Response to Reviewer 3SzE 4.0**
> > > >
> > > > The authors would like to thank the reviewer for their helpful discussion of the points that have been raised.
> > > >
> > > > The reviewer asks what the effect of a poor model estimate is on the quality of the nonlinear coherence estimation. To answer this, we first define $\delta$ such that $\hat{Y}=Y+\delta$ and then consider:
> > > > $$
> > > > J=E\left[|\hat{Y}-Y\|^2 \right]=E\left[\delta\delta^*\right]
> > > > $$
> > > > and
> > > > $$
> > > > ||\mathcal{F}_{\theta}(y+\delta)-x||^2 \ ,
> > > > $$
> > > >
> > > > which is minimised when minimising $\mathcal{L}^T_x$. The key assumption of the method is that minimising $\mathcal{L}^T_x$ is equivalent to minimising $J$, which depends on two factors. Firstly, in order to minimise $\mathcal{L}^T_x$, $\delta$ must be relatively small such that the argument of the reverse model is close to $y$. If $\delta$ is very large, this causes bias in the trained reverse model which reduces performance. This happens if both the model error and the level of additive noise are very high. Secondly,  both $y_{z}$ and $y_{n}$ must contain useful information to infer $x$, otherwise the method cannot provide a useful estimation of the nonlinear coherence: specifically $y_{z}$ must at least include a good estimate of the linear component of $y$, which is not difficult to achieve. This is because it needs to be driven to incorporate both signals.
> > > >
> > > > Additions to the limitations section (lines 509-522) aim to address this question.

---

> ### Comment · Reviewer_3SzE · 2024-11-28
>
> I thank the authors for their explanation. This seems like a big assumption, which I would think does not necessarily have to hold - given that $F$ is a CNN, its convolutional filters might average out some of $\delta$ so it doesn't affect the reconstruction of $x$. This again also seems to be an important logical step, which could have motivated and clarified the method, however the manuscript brushed over it.
>
> In conclusion, in my opinion both the method, and the insight obtained are hard to interpret, and I stick to my score.

---

### Author Response · Authors · 2024-11-23
**General response to reviews**

We thank the reviewers for their helpful, constructive comments. We have significantly revised the description of the method in order to improve clarity to a wider field. We have also emphasised the novelty and impact across a range of disciplines. We respond to your individual points below.

---

### Meta-Review · Area_Chair_fEYq · 2024-12-06

**Metareview:**

The authors aim in this work to provide a measure by which to judge the best possible dynamical system model that can fit experimentally measured inputs and outputs of a system. The measure focuses on a frequency based approximation of coherence,  and applies the metric to a number of systems.

There were a number of issues raised, primarily in the clarity and impact of the work. Multiple reviewers found the work difficult to parse, which clearly led to a number of confusions including in notation and in properly understanding the underlying assumptions the govern the work. Despite extended conversations with the reviewers, it seems that as it stands the assumptions tend to be limiting and the manuscript can be much more clearly honed to describe how the metric presented connects back to model goodness and interpretability. These seem to be significant drawbacks and a revised manuscript might fare better.

**Additional Comments On Reviewer Discussion:**

There was extended discussions, in particular with one reviewer, that ended up revealing much more about the underlying assumptions of the method. While benefiting the potential to clarify the paper's text, there was also the revelation of unclear assumptions that appear to be limiting. This discussion thus did not fully address all concerns despite significant effort on the part of the authors.

---

### Decision · Program_Chairs · 2025-01-22

Reject